# Local and Regional Anaesthetic Techniques in Canine Ovariectomy: A Review of the Literature and Technique Description

**DOI:** 10.3390/ani12151920

**Published:** 2022-07-27

**Authors:** Vincenzo Cicirelli, Matteo Burgio, Giovanni M. Lacalandra, Giulio G. Aiudi

**Affiliations:** Department of Veterinary Medicine, University of Bari “Aldo Moro”, 70021 Bari, Italy; matteo.burgio22@gmail.com (M.B.); giovannimichele.lacalandra@uniba.it (G.M.L.); giulioguido.aiudi@uniba.it (G.G.A.)

**Keywords:** local anaesthetic in canine ovariectomy, analgesia in canine neutering, regional techniques in canine ovariectomy

## Abstract

**Simple Summary:**

This review stems from a general trend of increasing attention toward surgical nociception in the veterinary field. Indeed, veterinarian anaesthetists are becoming more careful in relieving perioperative nociception, improving the analgesic protocols and therefore optimising surgical outcomes. Canine ovariectomy is a common surgical procedure with a moderate level of pain. Therefore, intraoperative analgesia is very important because pain causes various systemic inflammatory effects that slow patient recovery. This review aims to collect all recent information about local and regional anaesthetic/analgesic techniques in a review of the literature describing the technique utilised. In addition, the present review aims to provide practical guidelines for veterinary practitioners to improve the anaesthetic techniques in canine neutering through an overview of the available literature. In fact, the methods described, considering the relative simplicity of execution, can be used in daily routine practice.

**Abstract:**

Canine ovariectomy is an elective surgery with a moderate level of pain. Despite its relative simplicity, it requires surgical pain management. This study aimed to collect all recent information about local and regional anaesthetic/analgesic techniques in a review of the literature describing the technique utilised. The various procedures described in this review use local anaesthetics to improve analgesia in the routine systemic anaesthetic protocol. The approach described in this paper is called multimodal analgesia and is used in addition to the normal standard anaesthetic protocol. These techniques proved effective in minimising responses to the surgical stimulus and ensured adequate intraoperative and postoperative analgesia. The routine use of multimodal analgesia is considered a useful alternative for pain management in canine ovariectomy, in that it minimises patient suffering, improves the recovery of rescue analgesia, increases drug savings, and improves animal outcomes. In addition, the use of these local and regional techniques ensures satisfactory analgesic coverage that lasts for the first hours postoperatively.

## 1. Introduction

In recent decades, in veterinary medicine, the focus on the prevention, assessment, and treatment of surgical pain has substantially increased. As is known from the physiology of domestic animals, untreated pain results in many untoward responses involving all body systems: tachycardia, hypercoagulability, hypoventilation, hypoxemia, sepsis, stress, anxiety, reduced food intake, release of stress hormones, weight loss, immunosuppression, and increased blood pressure [1,2]. One of the objectives of veterinary medicine is to provide adequate analgesia to help the patient not feel pain and to move, eat, and sleep without discomfort, particularly in the first hours after the surgery [3]. Multimodal analgesia involves the use of multiple analgesic agents and provides the administration of both systemic and regional drugs. This method includes local and regional blocks that are safe and efficacious in dogs, when performed correctly, as described in this review. Local anaesthetic drugs can be injected directly into tissues to provide analgesia for manipulation or wounds or can be injected perineurally to provide analgesia for a wide variety of painful conditions. Owing to the potential to provide profound analgesia, these classes of drugs are recommended as part of the analgesic protocol in the majority of patients undergoing surgical procedures, such as canine ovariectomy. There are numerous local and regional blocks described for canine use, with the aim of reducing the dosage of the systemic drugs used and their possible side effects, while still maximising the desired effect [4]. Local anaesthetics have been used in analgesic protocols in the form of peripheral nerve blocks, epidural injections, intracavitary instillation, and regional blocks [5]. This review is a summary of the local and regional techniques known and used in ovariectomies of bitches as part of multimodal anaesthesia. This is a guide for practitioners to add to the classic anaesthetic protocol and locoregional techniques to improve analgesia in bitch neutering.

## 2. Local Anaesthetics

The mechanism of action of local anaesthetics is based on the blockade of sodium channels and the reversible blockage of the generation and propagation of electrical nerve impulses, which causes the blockage of the action of sensory and motor nerves. In veterinary medicine, local anaesthetics are extensively used in local and regional anaesthetic techniques. These techniques provoke desensitization of a localized area of the body, primarily causing ion channel blockers to act mainly on voltage-gated Na^+^ channels. The use of local anaesthesia results in the blockade of dependent K^+^ and Ca^2+^ channels, albeit with lower affinity [6,7,8,9,10], showing differential sensory and motor blocking behaviour; when this blockage is carried out at the level of the central neuraxis or peripheral nerves, it becomes clinically observable [11,12]. Loss of sensation regarding temperature, acute pain, light touch, and motor activity is among the actions of local anaesthetics [13]. Differential blockade is the name of this mechanism, and many factors influence it, such as the length of the nerve into which the anaesthetic will be inoculated, as well as the concentration and type of drug. After the administration of the anaesthetic drug, its availability is regulated by several concurring factors such as mass flow, diffusion, binding to neural and non-neural structures, and vascular absorption. The rate and amount of systemic absorption of these kinds of drugs should be taken into account due to the possibility of reaching toxic plasma concentrations. For this reason, local anaesthetics with low systemic absorption are more manageable. This is influenced by many factors such as the site of inoculation, the vasoactivity and lipid solubility of the drug itself, the dose used and additives (such as vasoconstrictors), vasodilation, i.e., the influence of local nerve block, and the physiological condition of the animal [14]. Lidocaine is used for infiltration anaesthesia, peripheral nerve block, epidural and intrathecal block, and regional intravenous anaesthesia. Its duration of action is around 1–2 h, but can be increased to 3 h by adding adrenaline (epinephrine) [15]. Mepivacaine is a similar drug, but has a somewhat longer duration of action due to its less vasodilatory properties. Due to its lower neurotoxicity than other local anaesthetics, it is widely used in equine medicine for peripheral diagnostic blocks [16]. Bupivacaine and levobupivacaine are strongly lipophilic molecules and are about four-times stronger than lidocaine; they have an onset of about 20–30 min and a longer effect duration of 3 to 10 h [17]. Bupivacaine has cardiotoxic properties in dogs (if injected IV at a dosage of 4–20 mg/kg); in fact, it is not recommended for intravenous regional anaesthesia, nor for topical anaesthesia. Its clinical use concerns the procedures of infiltrative, peripheral nerve, epidural, and intrathecal blocks. Ropivacaine has a structure and onset similar to bupivacaine and mepivacaine; they also have intrinsic properties of differential blocking, mainly at low concentrations; for this, they are very suitable when looking for a sensory block associated with a minimum motor dysfunction. This drug sees the same clinical use of bupivacaine and at equivalent doses is slightly less potent in motor blockade and determines a shorter marginal sensory blockade (6 h) [17]. Its biphasic effect means that at concentrations lower than 0.5% causes vasoconstriction, while at concentrations higher than 1%, it causes vasodilation [18]. Several substances can be combined with local anaesthetics to prolong the duration and increase the potency of nerve blocks. These include vasoconstrictors, corticosteroids, buffering substances, and alpha-2-agonists. The most common association is with adrenaline, which with its vasoconstriction effect decreases the absorption of the anaesthetic by the tissue vessels, thus prolonging the effect of the molecule and the duration of the blockade [19,20]. Sodium bicarbonate can increase the speed of onset and the potency of the anaesthetic used and is able to decrease the pain accompanying the injection [20,21,22]. In human medicine, it has been shown that administering local anaesthetics in combination with dexamethasone perineurally enhances and prolongs peripheral nerve block [23]. The mechanism of action is not yet known, but dexamethasone succeeds in decreasing the activity of C-type fibres [24]. Molecules belonging to the alpha-2-adrenergic class can decrease the clearance of local anaesthetics and have an inhibitory effect on C-fibres and A-delta fibres [25,26]. Their role is widely discussed in prolonging and intensifying regional anaesthetic blockade [27,28]. In addition, the pKa of different local anaesthetics affects their absorption differently. The alkalization of the local anaesthetic solution before it is injected, the speed of onset of the nerve block, and the alteration of tissue pH (e.g., inflammation, infections) can modify the correct absorption of the drug [21,22,29]. In general, LA can cause toxic effects in the nervous and cardiovascular systems. When a toxic dose is reached in plasma, inhibitory response mechanisms in the brain are blocked, leaving excitatory mechanisms free to act. This can lead to the onset of clinical signs such as muscular contractions and seizures. At the cardiac level, sodium channel blockade interferes with phase 0 of cardiac depolarization, the phase associated with the opening of the fast sodium channels. This mechanism is seen on the electrocardiogram (ECG) as an increase in PR and QRS intervals. It is known that the agents having high lipid solubility are more associated with toxicity, and R-enantiomers are more toxic than their laevorotatory counterparts. The system for treating signs of toxicity varies according to the severity. Convulsive activity can be kept under control with the use of benzodiazepines, and it is advisable to start supportive activity immediately with oxygen supplementation and, if necessary, mechanical ventilation. In the event of clinical signs suggestive of cardiovascular depression, action can be taken first with the use of fluids and cardiac inotropes. If arrhythmias occur, the use of other molecules acting as sodium channel blockers should be avoided. Local anaesthetics can also cause direct damage to tissues (neurotoxicity, chondrotoxicity), allergic reactions, and methemoglobinemia [30]. The toxic doses of the most commonly used LA reported in the literature for the canine species are lidocaine 20 mg/kg, bupivacaine 4.3 mg/kg, and mepivacaine 80 mg/kg [30].

### 2.1. Local Infiltration, Splash Block, and Injection into the Ovarian Pedicle

Carpenter et al. (2004) [31] stated that local anaesthetics should be used in reproductive system surgical procedures because the combination of local and general anaesthesia improves the analgesic effect, reducing intraoperative pain and the need for rescue analgesia [32,33]. The veterinary literature is in line with reports in humans, in which topical local and linear infiltration significantly reduce perioperative discomfort in patients undergoing laparoscopic and open ovariectomy [34,35]. During an obstetric surgery, the first surgical cut on the skin is the first moment related to nociceptive response. Somatic tissues can be infiltrated with local anaesthetic solutions to relieve pain associated with trauma and inflammation. This type of technique in open abdominal surgical approaches is commonly performed by subcutaneous injection of the linea alba [32]. Indeed, in both open and laparoscopic ovariectomies, ovarian manipulation is the most painful stage of surgery. The most studied technique is the “splash block”, which consists of irrigation of lidocaine into the peritoneum of the patient. The procedure consists of peritoneum lidocaine instillation during the ovariectomy. Lidocaine can also be infiltrated into the ovarian pedicle to obtain the analgesic effect before manipulation (Figure 1a,b). Following lidocaine application, surgical manipulation is discontinued for 90 s to allow the lidocaine to desensitise the fibres. In terms of mechanism, the local anaesthetic blocks the ascending afferent input, interfering with the ion channels of the nerves. Ovaries receive sympathetic fibres from the intermesenteric and caudal mesenteric plexus and parasympathetic fibres from the vagus nerve [36]. Lidocaine is absorbed quickly by ovarian tissue, blocking the ascending afferent input, interfering with the ion channels of the ovarian nerves, which receive sympathetic fibres from the intermesenteric and caudal mesenteric plexuses and parasympathetic fibres from the vagus nerve [37,38]. The current veterinary literature suggests that lidocaine splash provides intraoperative analgesia in dogs that have undergone OVH and video-assisted OVH [38]. Indeed, lidocaine has a fast onset (<2 min) and short duration of action (1–2 h); therefore, it is very useful in ovarian manipulation [39]. The splash block is also very safe, because the dose of lidocaine that produces central nervous system toxicity in dogs is 20.8 ± 4 mg/kg [31]. In this technique, the total dose of distilled lidocaine is 4 mg/kg administered over the duration of the surgical procedure; therefore, it is unlikely to be associated with side effects. Considering its relative simplicity, low cost, and safety, the splash block technique could be used in daily clinical practice. In addition to routine anaesthetic protocols, local anaesthesia is safe and does not cause cardiopulmonary suppression.

### 2.2. Tap Block

Transversus abdominis plane (TAP) block is an anaesthetic technique of great interest for locoregional anaesthesia in veterinary practice. This block is becoming increasingly popular in obstetric procedures with modest/high pain such as canine ovariectomy, ovariohysterectomy, and mastectomy. This technique involves the injection of a local anaesthetic into the TAP, obtaining its distribution over the thoracolumbar nerve branches located within this fascial plane. The lateral abdominal wall consists of three layers of muscles: the obliquus externus abdominis, obliquus internus abdominis, and transversus abdominis muscles. Between the obliquus internus abdominis and the transversus abdominis muscles, there is a fascial plane known as the transversus abdominis plane [40], where the ventral branches of the caudal intercostal (T9 to T12), costo-abdominal (T13), cranial iliohypogastric (L1), caudal iliohypogastric (L2), and ilioinguinal nerves (L3) are embedded. These nerves supply innervation to the abdominal muscles, subcutaneous tissue, mammary glands, abdominal skin, and underlying parietal peritoneum [41,42]. The contribution of the ventral branches of T7 and T8 to abdominal innervation is variable, and the ilioinguinal nerve does not reach the cutaneous ventral midline. A study evaluating TAP anatomy revealed that the ventral branches of the T10, T11, T12, costo-abdominal, cranial iliohypogastric, caudal iliohypogastric, and ilioinguinal nerves were located in the TAP in 100% of the abdominal walls evaluated, whereas the ventral branches of T7, T8, and T9 were present in 20%, 60%, and 95% of cases, respectively [40]. Several anatomical variations and different combinations of communicant branches contained in the TAP were found among the adjacent nerves from T7 to L3, forming a neural network called the TAP plexus [41]. The aim of the TAP block is to produce analgesia for surgical procedures performed on the abdominal wall, such as laparoscopic surgery, laparotomies and radicals, or partial mastectomies in dogs [43,44,45]. The TAP was previously described as a fascial plane located between the obliquus internus abdominis (OIA) and transversus abdominis (TA) muscles. At this point, nerve branches originate from the T11–T13 and L1–L3 spinal roots that innervate the abdominal part of the patient [46,47]. However, a precise description of the anatomical features of these nerve branches in dogs is limited. Various approaches to TAP block have been described. As these nerves are not typically visible on ultrasound, the injection end point for TAP block is the intermuscular fascial plane between the obliquus internus and the transversus abdominis muscles. Local anaesthetics injected within the TAP are assumed to reach the branches of the thoracolumbar spinal nerves, supplying sensory innervation to the abdominal wall [48]. Ultrasound-guided techniques have been described that allow visualisation of the layers of the abdominal wall, as well as visualisation of the needle and deposition of the local anaesthetic [49]. It has been suggested that ultrasound guidance of regional anaesthetic techniques may afford greater safety, efficacy, and efficiency while decreasing the incidence of local anaesthetic toxicity due to the visualisation of local anaesthetic spread, but clearly, it requires experience/training in ultrasound [50]. In veterinary medicine, ultrasound guidance offers the distinct advantage of directly visualising anatomic structures that may be poorly conserved among species or even breeds. In the case of this block, ultrasound guidance is essential to depositing local anaesthetic into the correct plane (Figure 2 and Figure 3). In a recent study by Romano et al. (2021) [48], two different techniques for injecting an anaesthetic solution were compared to include all the nerves in the anaesthetic block. The first technique used a lateral abdominal approach caudal to the last rib and cranial to the iliac crest, while the second used a subcostal approach caudal to the costal arch. The study, conducted on dog cadavers, showed that the best results, in terms of the area reached by the anaesthetic solution, were obtained with the subcostal technique. A combination of the two techniques is also proposed, using three injection points instead of a one, which also provides greater safety in terms of the possible toxic effects of local anaesthetic administration. The technique will need to be evaluated in vivo to assess its clinical implications as the literature indicates that in live animals undergoing celiotomy, there are factors such as respiration and lymphatic drainage that may influence the distribution of the drug and the systemic effect of the absorbed local anaesthetic [45,51,52,53].

### 2.3. Epidural Anaesthesia

Epidural anaesthesia is commonly used as part of a balanced anaesthetic protocol for perioperative pain management. This procedure is used in abdominal surgery and is particularly useful for the management of ovariectomy pain in dogs [54]. The advantage of using this route of administration is the possibility of blocking the modulation and transmission of nociceptive signals in the spinal cord, providing anaesthesia and analgesia of the dermatomes located in the caudal abdominal regions. Therefore, the epidural technique is particularly suitable for ovariectomy and ovariohysterectomy surgeries [55]. The meninges are composed of the dura mater, arachnoid membrane, and pia mater. The dura mater consists of a single meningeal layer and is separated from the vertebral periosteum by the epidural cavity, which is filled with fat and vertebral venous sinuses. Caudally, the spinal dura mater thins and forms part of the filum terminale. The subarachnoid cavity contains cerebrospinal fluid and is the space between the pia mater and the arachnoid membrane [56,57]. Epidural anaesthesia refers to the sensory, motor, and autonomic blocks produced by epidural administration of local anaesthetics. It is usually performed in the lumbosacral region in dogs. The volume of local anaesthetic injected usually ranges from 0.1 to 0.2 mL/kg [54]. The blockage area affects the caudal region of the umbilical scar (from L3 to S1), including the structures of the pelvic limb, perineum, and tail. The extent of cranial desensitization can vary depending on the anaesthetic volume, concentration, velocity of injection, amount of epidural fat, epidural space size, vascular absorption, and position of the patient. If a high injection volume is used (0.2 mL/kg), it can even reach T13 [32]. However, to obtain the desensitization of a more cranial area, greater volumes of local anaesthetic can be used, but it should be considered that this can induce a long-term blockade of sympathetic fibres, with consequent cardiovascular depression, peripheral vasodilation, hypotension, and bradycardia [58,59,60,61]. To perform epidural anaesthesia, it is necessary to locate the lumbosacral space (LS). It is recommended to place the anesthetized or sedated dog in sternal recumbency and keep it in this position for a few minutes after the injection to provide homogeneous and bilateral analgesia [57,62]. Usually, it is preferred to perform the epidural technique between L7 and S1 (LS) as the space between the two vertebrae is wider and access to the epidural space is easier to reach. However, if access to the lumbo-sacral space is not possible, the technique can be performed between L6 and L7 or in sacrococcygeal space [63]. To identify the point of insertion of the needle, it is advisable to move the hind limbs forward, so as to increase the space between L7 and the sacrum to facilitate needle access [56,61]. After performing hair trichotomy and disinfection of the area, it is possible to locate the LS by palpating both iliac crests with the thumb and middle finger. The imaginary line connecting them crosses the L6–L7 space. The index finger should be used to move caudally towards the sacrum. The LS appears as a small depression between the spinous process of L7 and sacral bone (Figure 4) [58,63,64]. Using a Tuohy needle, direct the bevel of the needle towards the animal’s head [62,65] and advance the needle at an angle of 45° from the vertical until an increase in resistance is felt, indicating puncture of the flavum ligament. If the needle enters the flavum ligament, a distinct “popping” sensation may be felt [66]; then, slowly advance, until the needle enters the epidural space (Figure 5). This can be identified using the following techniques: the loss of resistance (LOR), assessed using specific LOR syringes filled with air (due to the subatmospheric pressure present in the epidural space, the local anaesthetic will penetrate inside without any resistance). The second technique is the “hanging drop” technique (removing the stylet before the needle penetrates the flavum ligament and placing a drop of saline in the hub; once in the epidural space, the fluid is aspirated into the needle shaft by the subatmospheric epidural pressure) [58,65]. If cerebrospinal fluid is aspirated, the needle is mistakenly in the subarachnoid space. In this case, it is possible to go back and correctly reposition or perform subarachnoid anaesthesia, halving the dose of the local anaesthetic [56]. Another method is electrical nerve stimulation. This technique involves the use of an electrically isolated needle connected to the cathode of a nerve stimulator, inserted and advanced towards the epidural space. The anode is always positioned caudal to the cathode (e.g., on a limb). When the operator believes that the needle is correctly positioned, a current between 1 and 3 mA is applied. If the needle is placed in the epidural space, rhythmic muscle contractions of the pelvic limbs and tail should be evident [67,68]. To confirm the correct position of the needle, it is possible to perform epidurography (which involves inserting a contrast agent into the epidural space). Other useful techniques to identify the epidural space include the ultrasonograph-guided technique, fluoroscopy, or the observation of the epidural pressure waves (using a pressure transducer, connected to a monitor, able to measure epidural pressure waves, displayed on a monitor, and indicative of the correct positioning of the needle in the epidural space) [68,69]. The main contraindications to epidural anaesthesia use in dogs are infectious pathologies of the skin in the area of access of the needle, septicaemia, bacteraemia, skin trauma, and/or infections of the lumbosacral region. Patients with bleeding disorders, haemorrhage, and hypovolemic/hypotensive status should also be excluded. Trauma to the pelvic area, anatomical alterations, and obesity can make execution of the technique much more difficult [56,61,70]. The main complications that can occur during the execution of epidural anaesthesia are ineffective or partial block (frequent), the formation of haematomas or abscesses, nerve injuries, and incorrect intrathecal injections (occurs in 2–4% of cases during epidural anaesthesia in small animals) [56,58]. From a systemic point of view, it is possible to detect respiratory depression (if an excessive volume of local anaesthetic is used; this could migrate cranially up to the block of the nerves that control diaphragmatic activity), bradycardia and hypotension (sympathetic block), and neurological symptoms (coma, muscle twitching, and convulsions) [61,62]. Recently (although rare), cases of pruritus, lack of hair regrowth, myoclonus, and urinary retention have been reported [53,71,72]. The main local anaesthetics used in veterinary medicine for epidural anaesthesia are lidocaine, bupivacaine, and ropivacaine. Lidocaine has a shorter onset (5–15 min) and duration of action (60–75 min), while bupivacaine and ropivacaine have a slower onset (both 10–20 min), but a longer duration of action (>2 h), even if this depends greatly on the dosage used [62,63]. The duration of the desensitization produced by the epidural technique may be further influenced by the addition of adjuvant drugs. Among the most used are epinephrine (due to its vasoconstrictive effect, it is particularly useful in combination with lidocaine) and opioids (highly hydrophilic drugs, such as morphine, have a long duration of effect due to the delayed systemic absorption and longer retention in the spinal cord) [62]. Studies have shown a duration of action of morphine administered epidurally up to 12 h. However, for this reason, it is not a recommended drug in short-term surgeries such as castrations and ovariectomies [73,74,75]. Among the adjuvants currently available to increase the duration of action of epidural anaesthesia, there are also alpha-2-agonists and ketamine, which are still under study [76,77].

## 3. Conclusions

This review shows that the techniques and methods for controlling operative pain vary and are often new. In fact, in addition to the described techniques, quadratus lumborum (QL) block is being tested [78]. This is an ultrasound-guided technique of loco-regional anaesthesia involving the injection of a large volume of anaesthetic into the fascial plane surrounding the quadratus lumborum muscle, thus providing both somatic and visceral abdominal analgesia. QL block is an effective and safe method to provide intraoperative and postoperative analgesia for ovariohysterectomy in dogs, ensuring better pain control, and decreases the requirements of opioids in the perioperative period [78]. Regional anaesthesia has been increasingly used for analgesia in the perioperative period in veterinary medicine to improve pain control and the patient outcomes. Indeed, at this time, many studies are investigating inoculation to achieve inhibition of the largest number of nerves innervating the abdominal wall (T13-L7) [79,80]. This ongoing research on methods to improve analgesia in obstetric and gynaecological procedures in dogs shows that multimodal anaesthesia is a great approach. This method once again proves to be a better way of achieving anaesthesia using lower doses and combinations of effects, so that we can have a safer anaesthesia plan that can be controlled, but at the same time is pain-free and has fewer side effects. This work aims to show how many means are already available to practitioners, but at the same time, it shows how much more can be discovered by considering new perspectives and approaches.

## Figures and Tables

**Figure 1 animals-12-01920-f001:**
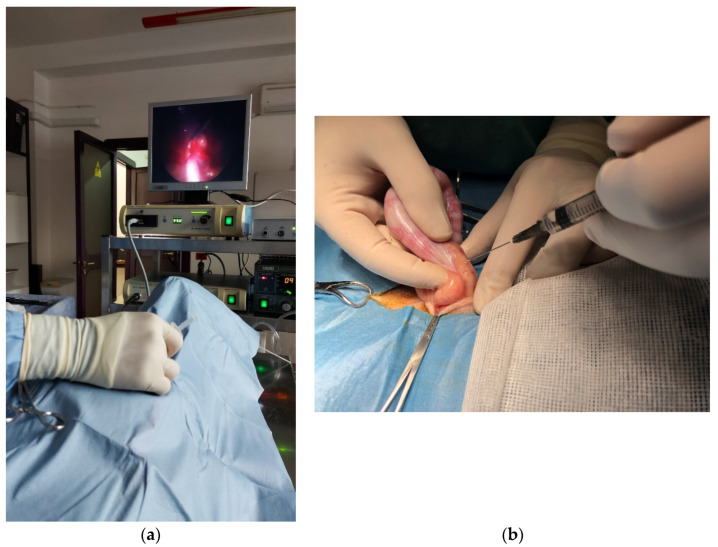
Lidocaine infiltration on ovarian pedicle during video-assisted OVH (**a**) and open surgery (**b**).

**Figure 2 animals-12-01920-f002:**
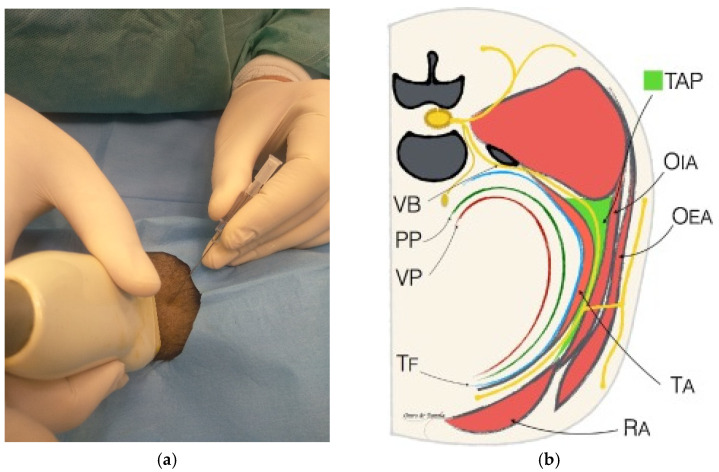
(**a**) Use of linear probe for the injection of a local anaesthetic into the TAP, obtaining its distribution over the thoracolumbar nerve branches located within this fascial plane. (**b**) Schematic illustration of a hemiabdomen showing the relationship of the abdominal nerves with the muscles of the abdominal wall. FT, transversalis fascia; OEA, obliquus externus abdominis muscle; OIA, obliquus internus abdominis muscle; PP, parietal peritoneum; RA, rectus abdominis muscle; TA, transversus abdominis muscle; TAP, transversus abdominis plane; VB, ventral branch of the spinal nerve; VP, visceral peritoneum [54].

**Figure 3 animals-12-01920-f003:**
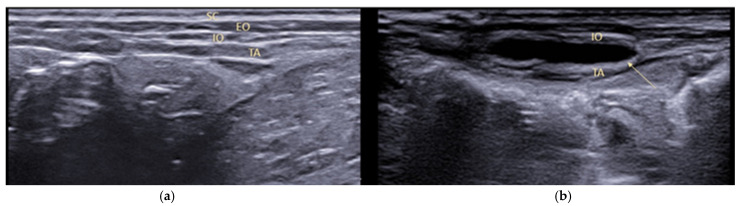
Ultrasonographic visualisation of the anatomical structures of the abdominal wall in a dog (SC, subcutaneous tissue; EO, obliquus externus; IO, obliquus internus; TA, transversus abdominis) before (**a**) and after (**b**) local anaesthetic injection. The patient in lateral decubitus.

**Figure 4 animals-12-01920-f004:**
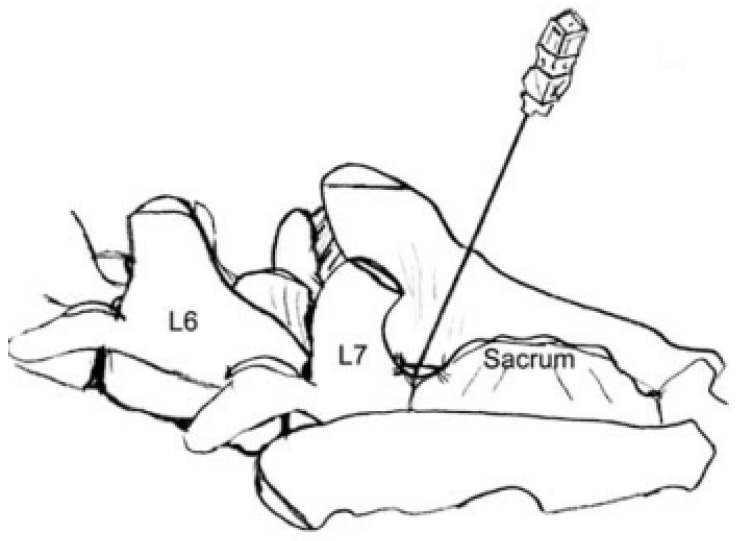
Anatomical position to identify the point of insertion of the needle [58].

**Figure 5 animals-12-01920-f005:**
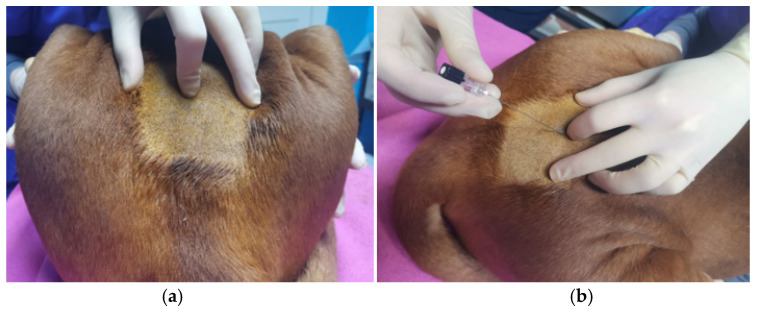
Identification (**a**) and insertion (**b**) of the needle into the LS.

## Data Availability

The data presented in this study are available on request from the corresponding author.

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
