# Peer review of "Local and Regional Anaesthetic Techniques in Canine Ovariectomy: A Review of the Literature and Technique Description"

_animals, 2022, doi:10.3390/ani12151920_

Round 1
Reviewer 1 Report
This is a nice review, but it would be helpful if you mention what the toxic dose is for each drug, for example you say bupivicaine is cardiotoxic, what is the toxic dose for the dog and cat? There are not the same. What are the toxic signs for the other drugs and what is their toxic dose. Also for epidurals most of us add morphine to the local anesthetic, is that worth mentioning?
Author Response
This is a nice review, but it would be helpful if you mention what the toxic dose is for each drug, for example you say bupivicaine is cardiotoxic, what is the toxic dose for the dog and cat? There are not the same. What are the toxic signs for the other drugs and what is their toxic dose. Also for epidurals most of us add morphine to the local anesthetic, is that worth mentioning?
Dear reviewer, first of all thank you for your appreciation of my work and also for your criticisms, which are useful for improving it. I corrected the paper and add same information in according to your suggestions. The revisions to the manuscript are marked up using the highlighter, such that you and the other reviewers can easily view any changes.
Reviewer 2 Report
Dear authors,
Thank you for submitting this work to Animals.
please see my comments in the attached file.
Best regards

Author Response
Summary:
Please use the terms “pain” properly – “nociception” would be a more appropriate term here.
Introduction:
there are many inaccuracies in this section
- the main mechanism of action of local anaesthetics is the blockade of sodium channels.
- The LA are not injected into the nerves but perineurally
- Lidocaine IS NOT the LA most widely used in modern anaesthesia
- There is limited evidence supporting the statement that corticosteroids enhance the
block duration or its intensity; in the clinical practise, they are usually combined to LA
for a different purpose (e.g., treating inflammation)
Please, revise.
- Dear reviewer, first of all thank you for your appreciation of my work and also for your criticisms, which are useful for improving it. I corrected the paper and add same information in according to your suggestions. The revisions to the manuscript are marked up using the highlighter, such that you and the other reviewers can easily view any changes.
TAP block
- “the anesthetic is then able to make contact with the thoracolumbar nerves”… Please
rephrase this (e.g., “the anaesthetic spreads…”
- Done
- I do not think there is any technique ever described for the TAP block which is not
ultrasound-guided. Please correct accordingly, or cite a proper reference.
- Regarding the conclusive statements, studies involving clinical patients have been
published; e.g., Portela DA, Romano M, Briganti A. Retrospective clinical evaluation
of ultrasound guided transverse abdominis plane block in dogs undergoing mastectomy.
Vet Anaesth Analg. 2014 May;41(3):319-24 (which also appears in your reference list).
- Done
- Please revise your reference list as some relevant references seem to be missing (e.g.,
Drożdżyńska M, Monticelli P, Neilson D, Viscasillas J. Ultrasound-guided subcostal
oblique transversus abdominis plane block in canine cadavers. Vet Anaesth Analg.
2017 Jan;44(1):183-186).
- Done
- The TAP block paragraph appears twice; please correct.
- I corrected the paper, add same information and references in according to your suggestions. The revisions to the manuscript are marked up using the highlighter, such that you and the other reviewers can easily view any changes.
Epidural
- If a high injection volume is used (0.2 ml/kg), it can even reach T13”: please reference
this sentence.
- Done
- “to move the desensitised area more cranially”: please reword as the area itself will not
be moved;
- Done
- “It is recommended to place the dog in sternal recumbency and keep it in this position
for a few minutes after the injection to provide homogeneous and bilateral analgesia”:
while this is relevant to spinal anaesthesia, it is less relevant for the epidural technique.
Please revise.
- Dear reviewer, thank you for your advice. However, the current literature and various authors affirm that this precaution should also be applied to the epidural. I am attaching here one of the references that demonstrated how post-epidural positioning alters the distribution of the anesthetic. Therefore, the authors chose to insert this statement as described by Gorgi et al and Steagall et al.
"Gorgi AA, Hofmeister EH, Higginbotham MJ, Kent M. Effect of body position on cranial migration of methylene blue injected epidurally in the lying position dogs. Am J Vet Res (2006) 67: 219–21. doi: 10.2460 / ajvr.67.2.219"
- Techniques for monitoring the needle advancement: there are more techniques than
those mentioned (e.g., pressure waves, epidurography, ultrasound-guided needle
insertion and so on). Please revise this part accordingly based on updated literature.
- Thank you for your suggestion. We have added additional methods as you requested.
- “However, these symptoms are mainly observed after intrathecal versus epidural
injections [61, 62]”: this is a very inaccurate statement as hypotension and bradycardia
are often seen also after epidural injection versus spinal technique.
- We change the phrase
- There are techniques missing (e.g., infiltration anaesthesia, injection in the ovarian ligament, splash blocks) that should be included.
- I have included the missing techniques
Conclusions
The conclusion should not be a summary of what already presented in the introduction. Please,
reformulate the conclusions paragraph.
- I modified the conclusion by mentioning the Quadratus Lumborum (QL) block, a technique still under study.
Reviewer 3 Report
This was a nice topic to write a review on.
The paper does not reflect the methods used everyday in a veterinary practice where speys are performed in dogs.
You should discuss the simplest methods of providing analgesia
1. Linear infiltration
2. Splash block (peritoneal)
3. Injection into the ovarian pedicle.
then discuss
4. TAP block (this would require the use of ultrasound and experienced user.
5. Epidural (with a local anaesthetic (discuss the peri and post operative consequences; and the use of other subsctances that could be injected into the epidural space e.g. preservative free opioids (morphine) with or without local anaesthetics). Ultrasound can also be used to locate the epidural space in obese animals. This too requires user experience.
6. Quadratus lumborumcblock. This is a speacialised block not routinely performed.

Author Response
This was a nice topic to write a review on.
The paper does not reflect the methods used everyday in a veterinary practice where speys are performed in dogs.
You should discuss the simplest methods of providing analgesia
- Linear infiltration
- Splash block (peritoneal)
- Injection into the ovarian pedicle.
then discuss
- TAP block (this would require the use of ultrasound and experienced user.
- Epidural (with a local anaesthetic (discuss the peri and post operative consequences; and the use of other subsctances that could be injected into the epidural space e.g. preservative free opioids (morphine) with or without local anaesthetics). Ultrasound can also be used to locate the epidural space in obese animals. This too requires user experience.
Dear reviewer, first of all thank you for your appreciation of my work and also for your criticisms, which are useful for improving it. I corrected the paper and add same information in according to your suggestions. The revisions to the manuscript are marked up using the highlighter, such that you and the other reviewers can easily view any changes.
Comment AH11: Again a diagram to illustrate anatomy might be suitable here? Done
Commented [AH12]: Why lumbosacral space? Why is that space safe in the dog? What happens if the lumbosacral space is not accessible? You could use sacrococcygeal space. Also, one can use an ultrasound to locate the epidural space in obese animals! Dear reviewer, thanks for your comments. We have increased the paragraph, in addition, we have implemented the discussions on epidural space identification techniques including neurostimulation and ultrasonography. Instead, as regards the sacrococcygeal epidural, it is a more suitable technique for soft tissue surgery of the perineum, tail and sacrum supplied by the pudendal, pelvic and caudal nerves (Grubb et al., 2020). In this review we have listed suitable techniques for the analgesic management of ovariohysterectomy, and, for which the sacrococcygeal has not been discussed.
Commented [AH13]: Is the animal sedated or anaesthetised or conscious? We have specified that the dog should be sedated or anesthetized.
Commented [AH14]: What about the type of local anaesthetic injected in the epidural space? Can it contain a preservative? Can you add something else to the local anaesthetic? Can you inject an opioid without injecting local anaesthetic? Done
Commented [AH15]: What about the duration of the epidural block with various local anaesthetics? Done
- Quadratus lumborumcblock. This is a speacialised block not routinely performed.
- I modified the conclusion mentioning the Quadratus Lumborum (QL) block, a technique still under study.

Round 2
Reviewer 3 Report
.Dear Authors,
Thank you very much for your hard work.
I am reading your paper as a veterinary anaesthetist.
There are a few questions which I have posed which need clarification! I have uploaded a Mcirosoft Word file with my suggestions.
I have requested the editors to seek a second opinion from another anaesthetist.
Best wishes

Author Response
Dear reviewer, thank you for sharing your valuable feedback and for your comments, which are useful in order to improve the manuscript.
- Every suggestion listed in PDF file was done. The revisions to the manuscript are marked up using the highlighter, such that you and the other reviewers can easily view any changes.
- I have followed all your advice and I have improved the local infiltration block section.
- In epidural paragraph we indicated the duration of different local anesthetics and we reported the possible side effects and risks of urinary retention. The aim of our review is to indicate the main techniques to be used during ovariectomy (specifying advantages and disadvantages). Our opinion is that the management of hospitalization and urinary retention is outside the scope of our work. In addition what is written in this paragraph what is described in the literatur. What is reported here corresponds to what was previously described by other authors in previous works for which it is necessary to indicate it.
- The conclusion was improved following your suggestions
